# The Adequacy of Current Legionnaires’ Disease Diagnostic Practices in Capturing the Epidemiology of Clinically Relevant *Legionella*: A Scoping Review

**DOI:** 10.3390/pathogens13100857

**Published:** 2024-10-01

**Authors:** Ryan Ha, Ashley Heilmann, Sylvain A. Lother, Christine Turenne, David Alexander, Yoav Keynan, Zulma Vanessa Rueda

**Affiliations:** 1Department of Medical Microbiology and Infectious Diseases, University of Manitoba, 745 Bannatyne Ave., Winnipeg, MB R3E 0J9, Canada; har3@myumanitoba.ca (R.H.); heilmana@myumanitoba.ca (A.H.); david.alexander@gov.mb.ca (D.A.); yoav.keynan@umanitoba.ca (Y.K.); 2Department of Internal Medicine, University of Manitoba, 750 Bannatyne Ave., Winnipeg, MB R3A 1R9, Canada; sylvain.lother@umanitoba.ca; 3Shared Health, Diagnostic Services, 1502-155 Carlton St, Winnipeg, MB R3C 3H8, Canada; cturenne@sharedhealthmb.ca; 4Cadham Provincial Laboratory, Shared Health, 750 William Ave., Winnipeg, MB R3E 3J7, Canada; 5Department of Community Health Sciences, University of Manitoba, 750 Bannatyne Ave., Winnipeg, MB R3E 0J9, Canada; 6School of Medicine, Universidad Pontificia Bolivariana, Circular 1ª 70-01, Barrio Laureles, Medellín 050031, Colombia

**Keywords:** pneumonia, *Legionella*, diagnostics, scoping review, epidemiology, incidence, Legionnaires’ disease, legionellosis

## Abstract

*Legionella* is an underdiagnosed and underreported etiology of pneumonia. *Legionella pneumophila* serogroup 1 (LpSG1) is thought to be the most common pathogenic subgroup. This assumption is based on the frequent use of a urinary antigen test (UAT), only capable of diagnosing LpSG1. We aimed to explore the frequency of *Legionella* infections in individuals diagnosed with pneumonia and the performance of diagnostic methods for detecting *Legionella* infections. We conducted a scoping review to answer the following questions: (1) “Does nucleic acid testing (NAT) increase the detection of non-*pneumophila* serogroup 1 *Legionella* compared to non-NAT?”; and (2) “Does being immunocompromised increase the frequency of pneumonia caused by non-*pneumophila* serogroup 1 *Legionella* compared to non-immunocompromised individuals with Legionnaires’ disease (LD)?”. Articles reporting various diagnostic methods (both NAT and non-NAT) for pneumonia were extracted from several databases. Of the 3449 articles obtained, 31 were included in our review. The most common species were found to be *L. pneumophila*, *L. longbeachae*, and unidentified *Legionella* species appearing in 1.4%, 0.9%, and 0.6% of pneumonia cases. Nearly 50% of cases were caused by unspecified species or serogroups not detected by the standard UAT. NAT-based techniques were more likely to detect *Legionella* than non-NAT-based techniques. The identification and detection of *Legionella* and serogroups other than serogroup 1 is hampered by a lack of application of broader pan-*Legionella* or pan-serogroup diagnostics.

## 1. Introduction

Legionnaires’ disease (LD) is caused by various species of *Legionella*, a genus of intracellular bacteria primarily found in water and soil. LD refers to pneumonia caused by *Legionella*. Legionellosis refers to *Legionella* infections, regardless of the site of infection. Within this text, Legionellosis is used when the source of the information uses the same terminology. Clinical symptoms vary from a mild febrile illness to severe and life-threatening pneumonia [1]. *Legionella* is suspected to be underdiagnosed in community-acquired pneumonia (CAP) due to non-specific presenting symptoms and signs, and thus there is an under-recognition by treating clinicians [2]. Pneumonia is commonly treated empirically without pathogen identification unless there is progressive clinical deterioration leading to further investigations [3]. *Legionella* are fastidious organisms that require special media for culture, and the sensitivity of culture techniques is low [4]. These limitations in LD detection and diagnostics have resulted in up to 90% of LD diagnoses being missed [5].

In 2022, the World Health Organization reported the overall mortality by Legionnaires’ disease as 5–10% and as high as 40–80% in immunosuppressed individuals [1]. The incidence of disease is estimated to be ten to fifteen cases per million per year in Europe, Australia, and the USA [1]. In 2019, the Public Health Agency of Canada stated that under 100 cases of LD were reported per year in Canada [6]. However, provincial data suggest a higher number of cases within Canada, with Ontario reporting over 100 cases per year from 2013 to 2022 [7]. Furthermore, LD cases have seen an increase across North America, as noted by the Centers for Disease Control and Prevention (CDC) and the British Columbia CDC [8,9].

Diagnostic testing for *Legionella* is rarely performed in mild CAP in community or hospitalized individuals and is only undertaken in severely ill hospitalized patients once other avenues have been exhausted. However, despite guidelines recommending that patients requiring hospitalization for CAP be tested for *Legionella* among other pathogens that may not be responsive to empirical therapy, the narrow spectrum of applicability, the limitations of most *Legionella* diagnostics, and the previously low rate of testing often make this effort fruitless [3,10,11]. The CDC and the National Collaborating Centre for Infectious Disease (NCCID) recommend diagnostic testing for *Legionella* in outpatients failing antibiotic therapy, individuals requiring intensive care admission, immunocompromised individuals, individuals with recent travel history, or in the setting of a known Legionellosis outbreak [4,8,12]. In addition, the CDC recommends testing for *Legionella* in cases of healthcare-associated pneumonia, and the NCCID recommends testing when there have been recent changes in water quality [12,13]. A recent review indicated a 42% increase in LD in the 3 weeks following storms [14]. LD is a looming hazard with the continuing rise in tropical storm intensity due to climate change, increased susceptibility associated with growing numbers of immunocompromised individuals, and a global aging population [14,15,16]. Additionally, the shortening of winter and the increase in average winter temperatures extends periods of rain and warm weather, further contributing to the risk of LD [14,15,17,18,19].

Out of the 65 species of *Legionella*, only 25 are known to cause disease in humans [20]. *L. pneumophila* is generally considered to be the most common cause of LD, followed by *L. longbeachae*, which is the cause of around 50% of LD cases in Oceania [21,22,23]. All clinically relevant species of *Legionella* are primarily found in water except the soilborne *L. longbeachae*. Effective treatments for LD include fluoroquinolones and macrolides [24]. Empirical therapy for CAP frequently includes either a macrolide or fluoroquinolones; however, when LD is not suspected and not treated, such as in immunocompromised people or people living with HIV, the outcomes and prognosis are adversely affected [25,26].

A summary of the diagnostic methods of LD is provided in Table 1. In most countries, the BinaxNOW urinary antigen test (UAT) remains the primary diagnostic test, with a high sensitivity and quick turnaround time for only a single serogroup of *L. pneumophila* (serogroup 1) [12,13,27,28]. In general, culture is the gold standard but has low sensitivity, requires invasive procedures to obtain samples (bronchoalveolar lavage) or involves samples that are infrequently produced in the context of LD (sputum), and offers no speciation in *Legionella* [12,29,30,31].

The prevalence of *Legionella*, especially non-*pneumophila* and non-serogroup 1 *pneumophila*, is likely underreported due to infrequent sampling, the unavailability of diagnostic tools in medical facilities, difficulty collecting sputum, long turnaround times, and the fact that it is only studied in severe clinical presentations or an outbreak context [24,27,32]. In this scoping review, we aimed to describe (1) the incidence, prevalence, and frequency of *Legionella* infections in individuals diagnosed with pneumonia with and without immunocompromised conditions, (2) the distribution of *Legionella* species and serotypes among people diagnosed with *Legionella*, and (3) which diagnostic techniques were used in each study.

## 2. Materials and Methods

### 2.1. Study Type

We conducted a scoping review following the scoping review checklist to answer the following questions [33]:Does nucleic acid testing (NAT) increase the detection of non-*pneumophila* serogroup 1 *Legionella* compared to non-NAT?Does immunocompromisation increase the frequency of pneumonia caused by non-*pneumophila* serogroup 1 *Legionella* compared to non-immunocompromised individuals with LD?

The population, intervention/exposure, comparator, outcome, and timeframe (PICOT) table for our questions are shown below in Table 2.

### 2.2. Search Strategy

We created a search strategy following PRISMA guidelines and searched in the following databases between 22 December 2022, and 12 February 2023: PubMed, PubMed Central, Cochrane Register of Controlled Trials, Clinicaltrials.org, and LegionellaDB. The full review protocol can be found in the Appendix A labeled as Appendix A. A timeframe was not included in the search. Our search queries, search strategies, and precise dates can be found in Appendix A. All searches were uploaded to Covidence [33], a web-based collaboration software platform that streamlines the production of systematic and other literature reviews [34].

### 2.3. Study Selection

We included studies that met all of the following criteria: (1) original research that reports data about the PICOT questions that can be used to calculate incidence, frequency, or prevalence of *Legionella* with a species or serogroup analysis; (2) comparative quantitation using multiple detection methods; (3) use of at least one NAT- and one non-NAT-based technique for diagnosis; (4) at least 5 cases of LD in the patient group. During the title and abstract screening, articles were included if there was an indication of clinical diagnosis of *Legionella*.

We excluded studies with the following criteria: (1) case reports or series of <5 patients; (2) studies missing serogroup or species analysis; (3) studies in which patients were only infected with *L. pneumophila* serogroup 1; (4) articles not available in English; (5) articles with no abstract; (6) articles examining environmental distribution; (7) ongoing trials; (8) studies not conducted on humans or using human samples; (9) non-pneumonic Legionellosis; (10) articles in which the diagnostic techniques are not reported.

Titles and abstracts were screened independently by two blinded reviewers using Covidence [34]. References that met all the inclusion criteria without exclusions were selected for full-text review and data extraction. Discrepancies were resolved by consensus or with a third reviewer where necessary.

### 2.4. Data Extraction

The following data were extracted from the included studies: (1) year of study; (2) country(ies) or continent where the study was conducted; (3) population under study; (4) sample size disaggregated by sex; (5) diagnostic techniques used; (6) brand of tests, if listed; (7) species or serogroups found; (8) immunosuppressive conditions; (9) comorbidities; (10) time of follow-up, if listed; (11) limitations, both recognized and unrecognized by original authors.

### 2.5. Data Synthesis

Descriptive analyses were conducted to report the frequency at which each serogroup or species was found clinically and the diagnostic testing used. Some studies were only eligible to describe the *Legionella* subgroup (i.e., species and/or serogroups) breakdown within a population, while other studies allowed for an analysis of the specific by-technique breakdown for each subgroup. Studies fulfilling the latter criterion were used to conduct an analysis on the testing positivity of different techniques. Furthermore, articles providing LD cases within a greater pneumonia context were subject to an analysis on their frequency within this context.

## 3. Results

Of the 3449 article citations identified, we included 279 unique studies for full-text review and 31 in the analysis (Figure 1). Many excluded articles were excluded for multiple reasons. However, we can only list one reason in Covidence.

Table 3 reports all the included articles, regions, populations studied, sample sizes, diagnostic test(s) used, and outcomes.

When looking strictly at studies reporting cases of pneumonia, regardless of pathogen identification, the majority of LD cases were found to be caused by unserogrouped *L. pneumophila*, followed by *L. longbeachae* and unidentified species of *Legionella* depending on the population included, with other identified species and serogroups being few and far between (Table 4 and Table 5). Relative to the total pneumonia cases, the most prevalent subgroups of *Legionella* are *L. pneumophila*, *L. longbeachae*, and unknown species of *Legionella* appearing in 1.4% (summed up across all serogroups), 0.895%, and 0.627% of the population, respectively (Table 4). However, a significant portion of these are due to unserogrouped *L. pneumophila* (1.140%) and unknown *Legionella* species (0.627%). For many subgroups, there is a wide range in their frequency of diagnosis.

Relative representation of subgroups relative to total reported LD cases is as follows: *Legionella pneumophila* serogroup 1, unserogrouped *L. pneumophila*, unspeciated *Legionella*, and *L. longbeachae* are the most common at 50.064%, 9.816%, 38.862%, and 0.482%, respectively (Table 5).

In studies involving national surveillance, the total population is reduced to either that of individuals diagnosed with pneumonia or LD, depending on the information provided in the study. Subgroups examined in only one eligible study have their frequencies marked with an asterisk (*) (Table 4 and Table 5).

Seven studies qualified for our technical analysis, which required a by-technique breakdown, including the reporting of all techniques performed on each sample instead of only listing the main technique used for diagnosis (Table 6).

Additionally, studies were unclear about whether testing using the UAT was attempted for initial diagnoses of cases later found to be compatible with non-*pneumophila* serogroup 1 *Legionella* but not LpSG1. Where it would normally be expected to give a negative result, it is unclear if the UAT was used at all, as the studies did not report its use either way.

## 4. Discussion

Our review found that most LD cases are caused by an unidentified species or serogroup of *L. pneumophila*. The emphasis on using a UAT that strictly detects LpSG1 as an initial test likely results in a significant number of missed cases [32,62,63]. Typically, culture-positive diagnoses are not subject to speciation or serogrouping or the species could not be identified for other reasons [36]. We found that almost 50% of LD cases are still caused by an unspecified species or a serogroup not detected by the standard UAT (Table 5). In general, the detection of the respiratory pathogen causing CAP is very low. Jain et al. found that only 38% of individuals with CAP were positive for pathogen detection [64].

The continued belief that LD is almost exclusively caused by *L. pneumophila* SG1 and the accompanying diagnostic practices leads to further missed diagnoses and increasingly discordant epidemiological data. In addition, the broad diagnostic coverage of *Legionella* is crucial, as the disease has been reported to be similar in both manifestations and outcomes across subgroups [38,65,66]. LD diagnosis is very narrow in scope and species identification is often not involved, further reinforcing the small pool of clinically relevant *Legionella*. While the BinaxNOW *Legionella* UAT has a rapid turnaround time, it is strictly capable of detecting *L. pneumophila* serogroup 1. The RIBOTEST *Legionella*, another UAT, is used exclusively in Japan and can detect *Legionella pneumophila* serogroups 1–15 and several other species at a sensitivity comparable to BinaxNOW *Legionella*’s rate of detection for LPSG1 strains suspended in saline solutions [67]. The other first-line diagnostic is culture, which is hampered by low sensitivity, takes several days, and has variable performance between laboratories. One potential hurdle to culture usage beyond the slow growth rate of *Legionella* species is the use of BCYE media without antibiotics, allowing the growth of other bacteria. Some newer products have been released on the market that may improve the landscape of *Legionella* diagnostics. Among these is the BioFire^®^ FilmArray^®^ Pneumonia (PN) Panel, which can detect up to 33 bacterial and viral targets (including *Legionella pneumophila*) in bronchoalveolar lavage fluid or sputum [68].

Based on our dataset, we found that NAT-based techniques had a higher frequency of *Legionella* positivity compared to non-NAT-based ones (Table 6). PCR remains unstandardized across institutions, as many groups use in-house primers and divergent protocols. Ideally, PCR should capture a diverse population of *Legionella* by using a multiplex panel of primers targeting genus-conserved sequences (e.g., 16S rRNA) and species-conserved sequences in some of the more common species of *Legionella* such as *L. pneumophila* and *L. longbeachae*, with some adjustments made based on locality. While costly, the adoption of sequencing into diagnostics would contribute to better surveillance of clinically relevant strains, changes in drug resistance, and mutations over time [69]. Of note, European countries have shifted towards sequence-based typing for *Legionella* rather than serogroup, which can be used to obtain sequences and typify from culture-negative samples [70].

One of the key issues in pneumonia treatment is that the etiology of CAP is rarely identified, and empirical treatments may not cover atypical bacteria, especially within the context of people living with HIV (PLHIV) [3,71]. In most scenarios, the American Thoracic Society and Infectious Diseases Society of America do not recommend diagnosis until initial therapies have been exhausted, even suggesting the aversion of the UAT, unless the disease is severe [3,72]. *Legionella* cases requiring ICU admissions are associated with delayed urinary antigen testing and presumably *Legionella* testing in general [73]. Conversely, early concordant treatment reduces the probability of ICU admission [25,73]. Li et al. found that next-generation sequencing effectively detected fastidious organisms including *Legionella* in cases that culture failed to identify, even detecting pathogens when culture test results were negative, resulting in adjusted treatment in 55% of patients [74]. Thus, the delays and mistakes in *Legionella* identification are likely contributing to the high number of cases requiring hospitalization and intensive care.

Presumably, the reported case numbers of non-LpSG1 are inaccurate due to missed diagnoses. This is a conceivable scenario, as there are regional variations in species distribution and unidentified species were the third most common finding in our review [75]. Furthermore, many studies use culture or UAT for the initial diagnosis before further identification, only testing samples with other techniques when initial tests showed positive. Necessitating a positive result from these techniques introduces a bias toward the detected species or serogroups. Samples for culture are rare due to difficulties in obtaining bronchoalveolar lavage and sputum, while the UAT is over-selective [13,27,32,74]. Our findings show that the subgroups that warrant more deliberate monitoring are *L. pneumophila* regardless of serogroup, *L. longbeachae*, *L. micdadei*, and *L. bozemanae* (Table 4 and Table 5).

While it has been widely recognized that diagnosis of *Legionella* pneumonia is poor, Table 6 highlights the need for diagnostics to be more comprehensive. While accommodating the detection of all species of *Legionella*, the inter-study and presumably inter-lab consistency of PCR is unknown. The information included in our dataset suggests that PCR and DNA probes had a higher frequency of positivity, regardless of *Legionella* subgroup (Table 6). Of note, DNA probes were only used in one study, and of the studies that reported which type of PCR was used, all used real-time PCR. The difficulty in recovering respiratory samples further encourages the merits of testing multiple specimens. Without having to develop new techniques or products to detect *Legionella* in patient samples, there is a benefit in adopting a more comprehensive diagnostic regimen as conducted by Pasculle et al. and Decker et al. [24,53,73,76]. However, these are not without the drawback of a higher rate of false positives.

Beyond the consequences of a delayed diagnosis and impeded timely concordant treatment, limitations of *Legionella* diagnostics exacerbate the issue further by also delaying the recognition of outbreaks [5,77,78,79,80]. Identifying an outbreak of *Legionella* is crucial, as typical outbreak strategies such as isolation of cases will not reduce cases, and the medium by which *Legionella* is contracted (water) makes outbreaks very likely. The delay also contributes to increasing the mortality rate, as in immunosuppressed individuals the mortality rate can be up to 80% in infections across a range of different species of *Legionella* if left untreated [5,81]. Furthermore, up to 90% of *Legionella* infections are missed even in environments that are equipped for diagnosis and the increasing mortality rate, warranting an increase in testing frequency [5]. Our findings are in agreement with Decker et al., who found that systematically testing for *Legionella* in people diagnosed with pneumonia using two diagnostic techniques indicated that *Legionella* diagnoses had been underreported [76].

An opportunity for further research is to determine if PLHIV or other forms of immunosuppression have a higher susceptibility to specific subgroups of *Legionella*. While we had sought to investigate this, the lack of reporting of *Legionella* cases impeded our efforts in doing so. Head et al. found that 36% of PLHIV were coinfected with *Legionella* [81]. In their study, approximately one-third of the *Legionella* infections were caused by *L. pneumophila*, none of which were LPSG1 [81]. However, it is unclear if these proportions are due to geographical factors or the presence of HIV. Sivagnanam et al. found that 31% and 47% of their American transplant recipient cohort were infected with *L. micdadei* and *L. pneumophila*, respectively [60]. These studies emphasize the need to diversify diagnostic methods, especially in the immunocompromised population.

A limitation of our study is that further analyses cannot be performed because of a lack of information regarding the specific tests used, the differences in populations tested, the sample storage conditions, and that patients received inconsistent testing. Furthermore, studies used different techniques as their standards, only subjecting samples to further diagnosis with an initial positive result. Despite the heterogeneity in testing protocols and specific primers, PCR provided positive results in most tests. We were also unable to draw any conclusions about the effect of geographical distribution and regional testing conventions with the number of studies included in our review. Additionally, researchers are more likely to publish results that reveal particularly high or low prevalence, which may cause intermediate cases to be unrepresented within the literature. This was also the reasoning behind our inclusion criteria “at least 5 cases of LD in the patient group”, as articles reporting very few cases tended to be case reports/series or looking at isolated cases of LD without the generalized pneumonia context. Additionally, among the 30 studies that were excluded for the insufficient presence of *Legionella*, 15 found 0 cases of LD and 2 studies found 4 cases. The remaining 13 studies were also excluded for additional reasons, most frequently a lack of species/serogroup analysis and/or the absence of an NAT-based diagnosis.

In conclusion, the real epidemiology of *Legionella* infections is unclear due to the lack of an adequate diagnostic test that identifies other non-*pneumophila* serogroup 1 *Legionella* and different criteria on who, when, and how to diagnose *Legionella* [22,23,29,32,80]. It is essential to isolate strains and carry out epidemiological research studies using whole-genome sequencing to identify and track circulating strains of *Legionella* for future diagnostic test development, strains of concern, and clinical guideline updates.

## Figures and Tables

**Figure 1 pathogens-13-00857-f001:**
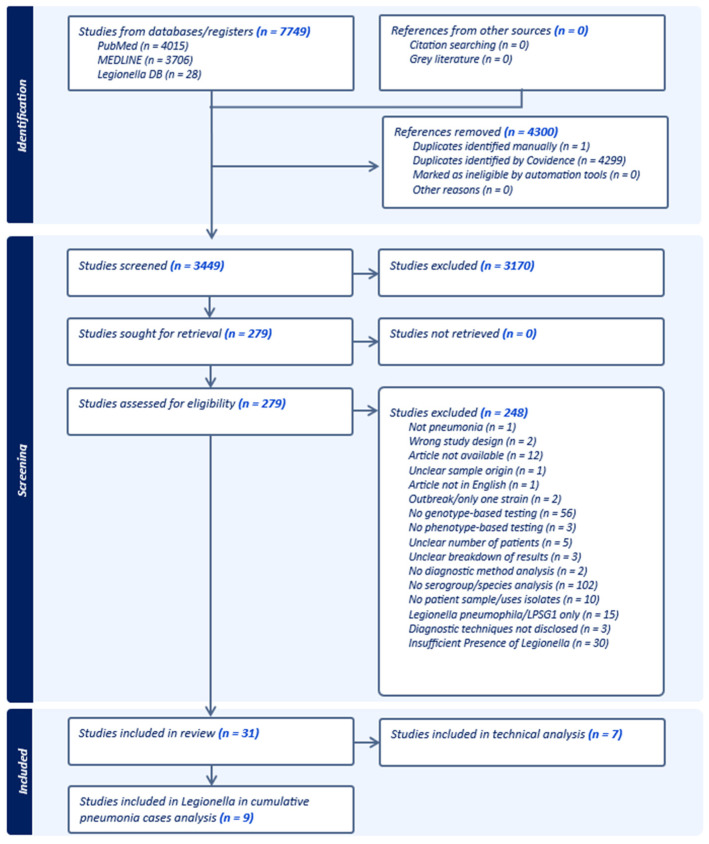
PRISMA flowchart depicting the screening process generated by Covidence [34].

**Table 1 pathogens-13-00857-t001:** Current *Legionella* diagnostic techniques outlined by the Centers for Disease Control and Prevention (CDC) (4) ^a^.

Test	Sensitivity (%)	Specificity (%)	Advantages	Disadvantages
Culture	20–80	100	Detects all species/serogroups	Technically difficult
				Slow (3–5 days to grow)
				Sensitivity dependent on technical skill Affected by appropriate antibiotic therapy No species identification without further testing
Urinary antigen test	70–100	95–100	Rapid	Only detects *L. pneumophila* serogroup 1 (LpSG1)
			Non-invasive	Some patients do not excrete the antigen or excrete the antigen intermittently
Polymerase chain reaction (PCR)	95–99	>99	Rapid	Influenced by specimen quality
			Can detect species/serogroups other than LpSG1	Assays vary by laboratory
				Limited commercial availability
Direct fluorescent antibody	25–75	>95	Can detect species/serogroups other than LpSG1	Technically difficult
				Reagents may be difficult to obtain
Serology	80–90	>99	Can detect species/serogroups other than LpSG1	Antibodies may be shared across species/serogroups Cannot distinguish between current and past infection

^a^ Modified from tables from the CDC [4].

**Table 2 pathogens-13-00857-t002:** PICOT table outlining our research questions.

Scoping Review Question	Population	Intervention/Exposure (Hypothesis)	Comparator	Outcome	Time Frame
1	Individuals with pneumonia	Genotype-based techniques such as PCR and sequencing	Phenotype-based techniques such as culture, serology, DFA, and UAT	Incidence, prevalence, and frequency of *Legionella* compatibility (specific species/strains as stratified by molecular techniques)	N/A
2	Individuals with pneumonia who are immunocompromised	Genotype-based techniques such as PCR and sequencing	Phenotype-based techniques such as culture, serology, DFA, and UAT	Incidence, prevalence, and frequency of *Legionella* compatibility (specific species/strains as stratified by molecular techniques)	N/A

**Table 3 pathogens-13-00857-t003:** Summary of studies included in the scoping review.

First Author, Year of Publication, Reference	Region(s)	Year(s) of Study	Population	Sample Size (Cases)	Sample Types	Techniques Used	*Legionella* Species or Serogroups Found ^b^
Alexiou-Daniel et al., 1998 [35]	Greece	1993–1998	Hospitalized legionellosis patients	24	Serum	Serology ^a^, culture, and PCR	22 LpSG1, 2 LpSG4
Beauté, 2017 [36]	European Union states, Iceland, Norway	2011–2015	*Legionella*-infected individuals	30,532	Urine and serum	UAT, culture, and PCR	3020 LpSG1, 19 LpSG2, 101 LpSG3, 13 LpSG4, 19LpSG5, 42 LpSG6, 9 LpSG7, 8 LpSG8, 5 LpSG9, 19 LpSG10, 3 LpSG11, 1 LpSG12, 2 LpSG13, 7 LpSG14, 4 mixed SG, 7 non-LpSG1, 232 Lp unknown SG, 2 La, 15 Lb, 1 Lc, 2 Ldu, 35 Ll, 1 Lma, 12 Lmi, 1 Ls, 27 other *Legionella* species, and 38 unknown *Legionella* species
Berger et al., 2006 [37]	France	2002–2003	ICU pneumonia	210	BALF, serum, and urine	RT-PCR, UAT, serology, and culture	10 Lp, 3 Lb, 2 La, 2 Lr, 1 Lq, and 1 Lwo
Cameron, 2016 [38]	Scotland	2008–2014	*Legionella*-infected individuals	37	Respiratory samples, sputum, and urine	PCR, serology, culture, and UAT	12 Ll and 25 Lp
Diederen et al., 2008 [39]	Netherlands	1998–2000	CAP adults	242	Sputum, endotracheal aspirates, lung biopsy, and bronchoscopic specimens	RT-PCR, UAT, and blood culture	11Lp and 2 non-pneumophila *Legionella*
Diederen et al., 2009 [40]	Netherlands	2002–2005	Pneumonia compatibility	151	Sputum, blood, urine, serum, and BALF	RT-PCR, culture, UAT, and ELISA	36 Lp and 4 non-pneumophila *Legionella*
Elverdal et al., 2013 [41]	Denmark	2008–2010	Pneumonia (hospitalized)	10,503	Serum, LRT samples, blood, and urine	ELISA, culture, UAT, and PCR	35 LpSG1, 1 LpSG2, 11 LpSG3, 1 LpSG5, 3 LpSG6, and 62 *Legionella* spp.
Ghorbani et al., 2021 [42]	Iran	2019–2020	Pneumonia (hospitalized)	123	Sputum, BALF, and pleural aspirates	Culture and RT-PCR	8 Lp and 1 Lc
Isenman et al., 2016 [43]	New Zealand	2009–2013	*Legionella*-infected individuals	126	Urine, LRT samples, and serum	Culture, PCR, UAT, and serology	107 Ll and 19 Lp
Jespersen et al., 2009 [44]	Denmark	1995–2005	*Legionella*-infected individuals	370	Urine and serum	UAT, serology, PCR, and culture	110 LpSG1, 4 LpSG2, 39 LpSG3, 3 LpSG4, 6 LpSG6, 4 Lp unknown serogroup, 4 Lb, 2 Lmi, and 161 unknown *Legionella* species
de Jong et al., 2010 [19]	Europe	2010	*Legionella*-infected travelers	864	Urine and serum	UAT, PCR, culture, and serology	672 LpSG1, 3 LpSG3, 2 LpSG6, 1 LpSG12, 3 mixed SG, 158 unknown SG Lp, 1 Lb, 10 unknown species, and 14 unreported species
Joseph, 2004 [45]	Europe	2000–2002	*Legionella*-infected travelers	10,322	Urine and serum	UAT, serology, PCR, direct antigen, and culture	7900 LpSG1, 1749 Lp, 9 LpSG2, 35 LpSG3, 5 LpSG4, 10 LpSG5, 22 LpSG6, 1 LpSG7, 2 LpSG8, 7 LpSG10/14, 673 non-*pneumophila Legionella*, 2 La, 4 Lb, 2 Ldu, 1 Lg, and 3 Ll
Joseph et al., 2010 [46]	Europe	2007–2008	*Legionella*-infected individuals	11,867	Serum, urine, and respiratory samples	Culture, UAT, serology, respiratory antigen, and PCR	9436 LpSG1, 1785 non-SG1 Lp, and 646 unknown/other species
Kim et al., 2015, [47]	South Korea	2000–2001	Suspected LD	10	Sputum and urine	UAT, serology, RT-PCR, and culture	5 LpSG1, 1 LpSG 2-14, and 4 unknown *Legionella* spp.
Lever, 2003, [48]	Europe	2000–2001	*Legionella*-infected travelers	841	Urine, serum, and respiratory samples	Urine, serology, PCR, and culture	303 LpSG1 and 407 other serogroup/species
Lindsay et al., 1994 [49]	Scotland	N/A	Proven cases of LD	5	Serum and urine	UAT, serology, culture, and PCR	4 LpSG1 and 1 LpSG12
Löf et al., 2021 [50]	Sweden	2018	Non-*pneumophila Legionella* cases	41	N/A	UAT, RT-PCR, culture, and serology	6 non-*pneumophila Legionella*, 33 Ll, and 2 Lb
Maniwa et al., 2006 [51]	Japan	1999–2005	*Legionella*-infected individuals	30	Urine, sputum, BALF, and serum	culture, UAT, PCR, and serology	10 LpSG1, 2 LpSG6, 1 Ll, and 17 unknown *Legionella* species
Murdoch et al., 1996 [52]	New Zealand	1992–1995	Previously confirmed LD positive or negative	52	Urine and serum	PCR, culture, serology, and ELISA	2 LpSG1, 1 LpSG3, 3 LpSG4, 2 LpSG5, 1 LpSG6, 1 LpSG7, 1 LpSG10, 2 LpSG12, 2 LpSG13, 10 Lmi, 3 Ll, 3Lj, 1 Lb, and 1 Lg
Pasculle et al., 1989 [53]	US	1987	*Legionella*-infected individuals	809	Sputum	ELISA, culture, and DNA probe	6 LpSG1, 2 LpSG4, 1 LpSG6, 1 Lmi, and 2 LpSG1/Lmi
Pouderoux et al., 2019 [54]	France	2013–2017	Culture-positive LD	1686	Sputum, bronchial aspirate, and BALF	Culture, real-time RT-PCR, WGS, and serology	9 LpSG1, 1 LpSG3, 1 LpSG8, 1 *Legionella* spp., and 1 LpSG2/6/12
Priest et al., 2019 [55]	New Zealand	2015–2016	Pneumonia	4826	LRT specimens, and urine	PCR, culture, MALDI-TOF, serology, and UAT	52 Lp, 150 Ll, 24 other *Legionella* species, and 12 non-speciated *Legionella*
Qin et al., 2016 [56]	China	2012–2013	Pneumonia or LRTIs in hospital	624	BALF and sputum	Culture, RT-PCR, and sequencing	70 Lp and 1 other *Legionella* species
Ricketts et al., 2005 [57]	Europe	2003–2004	*Legionella*-infected individuals	9166	Urine, serum, and respiratory samples	Culture, UAT, serology, antigen detection, and PCR	7007 LpSG1, 1526 non-SG1 Lp, and 633 other *Legionella* spp.
Ricketts et al., 2007 [46]	Europe	2005–2006	*Legionella*-infected individuals	11,980	Respiratory samples, urine, and serum	Culture, UAT, serology, and PCR	9219 LpSG1, 1862 Lp non-SG1/unknown SG, and 899 unknown *Legionella* species
Ricketts et al., 2010 [58]	Europe	2008	*Legionella*-infected individuals	866	N/A	Culture, serology, PCR, and UAT	57 LpSG1, 1 LpSG2, 1 LpSG3, 3 Lp unknown SG, and 1 unknown *Legionella* spp.
Scaturro et al., 2021 [59]	Italy	2018	Pneumonia patients	33	Urine, serum, and respiratory secretions	UAT, RT-PCR, culture, serology, and single high titer	18 Lp unclear SG and 15 LpSG2
Sivagnanam et al., 2017 [60]	US	1999–2013	Transplant recipients suspected of *Legionella* infection	4090	BALF, blood, sputum, urine, and tissue	Culture, UAT, sequencing, and MALDI-TOF	7 LpSG1, 8 unknown SG Lp, 10 Lmi, 4 Ll, 1 Lwa, 1 Lt, and 1 Ldu
Tateda et al., 1998 [61]	Japan	N/A	Suspected *Legionella* infection	36	Sputum, BALF, serum, and urine	Culture, serology, UAT, and PCR	12 Lp, 1 Lb, and 1 Lp/Ldu
Waller et al., 2022 [21]	Australia	2010–2021	Legionellosis patients	53	Urine and serum	UAT, serology, and PCR	31 Ll, 22 Lp, and 2 *Legionella* spp.

ELISA includes DFA. Unspecified PCR techniques are listed as “PCR”. Abbreviations: LP, *L. pneumophila*; Lb, *L. bozemanii* or *L. bozemanae* (*sic*); La, *L. anisa*; Lr, *L. rubrilucens*; Lq, *L. quinlivanii*; Lwo, *L. worsleiensis*; Lc, *L. cherrii*; Ll, *L. longbeachae*; SG, serogroup; Ldu, *L. dumoffii*; Lmi, *L. micdadei*; Lj, *L. jordanis*; Lg, *L. gormanii*; Lwa, *L. wadsworthii*; Lt, *L. tucsonensis*; Lc, *L. cincinnatiensis*; Lma, *L. maceachernii*; Ls, *L. sainthelensi*; BALF, bronchoalveolar lavage fluid; LRT, lower respiratory tract; PCR, polymerase chain reaction; UAT, urinary antigen test; ELISA, enzyme-linked immunosorbent assay; WGS, whole-genome sequencing; SBT, sequence-based typing; RT, real-time. ^a^ Serology includes seroconversion and high antibody titers against *Legionella*. ^b^ Species separated by a slash (“/”) are present as coinfections.

**Table 4 pathogens-13-00857-t004:** Breakdown of reported cases of *Legionella* subgroups relative to the cumulative population of pneumonia cases ^a^.

Species	People Diagnosed with *Legionella*	Total Population Diagnosed with Pneumonia	Frequency	Frequency Range
LpSG1	35	16,751	0.209%	0.333–23.8%
LpSG2	1		0.006%	0.0095% *
LpSG3	11		0.066%	0.105% *
LpSG5	1		0.006%	0.0095% *
LpSG6	3		0.018%	0.00286% *
Lp unknown SG	199		1.190%	1.07–33.3%
*L. longbeachae*	150		0.895%	0.0978–3.085%
*L. bozemanae*	4		0.024%	1.43–2.778%
*L. anisa*	2		0.012%	0.952% *
*L. rubrilucens*	2		0.012%	0.952% *
*L.* *cherrii*	1		0.006%	0.813% *
*L.* *dumoffii*	1		0.006%	2.778% *
*L.* *quinlivanii*	1		0.006%	0.476% *
*L.* *worsleiensis*	1		0.006%	0.476% *
Unknown *Legionella* spp.	105		0.627%	0.160–2.649%
Total	509		3.086%	-

^a^ Data compiled from 9 studies. Abbreviation(s): Lp; *L. pneumophila*. ^a^ In studies involving national surveillance, the total population is reduced to that of individuals diagnosed with pneumonia based on the information provided in the study. Studies only reporting patients who were positive for Legionnaires’ disease were excluded from the analysis summarized in this table. Total population includes individuals who presented with pneumonia but tested negative for *Legionella* and those who tested positive. Frequency ranges show the frequency of *Legionella* within the population of a given study. Serogroups and species examined in only one eligible study for this analysis have their frequencies marked with an asterisk (*).

**Table 5 pathogens-13-00857-t005:** Relative frequency of *Legionella* serogroups and species compared to total *Legionella*-positive cases in the literature ^a^.

Serogroups	Number of Cases	Frequency
LpSG1	38,506	50.064%
Lp unknown serogroup	7550	9.816%
LpSG3	194	0.252%
LpSG6	79	0.103%
LpSG2	51	0.066%
LpSG5	32	0.042%
LpSG4	28	0.036%
LpSG10	27	0.035%
LpSG14	14	0.018%
LpSG7	11	0.014%
LpSG8	11	0.014%
LpSG9	7	0.008%
LpSG12	6	0.008%
LpSG13	4	0.005%
LpSG11	3 *	0.004%
**Species**		
Unknown *Legionella* spp.	29,890	38.862%
*L. longbeachae*	371	0.482%
*L. micdadei*	37	0.048%
Unknown non-*pneumophila Legionella*	37	0.048%
*L. bozemanae*	31	0.040%
*L. anisa*	6	0.008%
*L. dumoffi*	6	0.008%
*L. gormanii*	3	0.004%
*L. maceachernii*	2	0.003%
*L. rubrilucens*	2 *	0.003%
*L. quinlivanii*	1 *	0.001%
*L. worsleiensis*	1 *	0.001%
*L. cherrii*	1 *	0.001%
*L. wadsworthii*	1 *	0.001%
*L. tucsonensis*	1 *	0.001%
*L. cincinnatiensis*	1 *	0.001%
*L. sainthelensi*	1 *	0.001%

^a^ Serogroups and species examined in only one eligible study have their frequencies marked with an asterisk (*). Abbreviations: Lp, *L. pneumophila*; SG, serogroup. Data compiled from 31 studies.

**Table 6 pathogens-13-00857-t006:** Qualitative summary of diagnostics by *Legionella* species and serogroup, separated by technique ^a^.

First Author	Spp/SG	UAT	PCR	Serology	DFA	Culture	DNA Probe
Ghorbani [42]	*L. pneumophila*	-	7/7	-	-	2/7	-
	*L. cherrii*	-	1/1	-	-	0/1	-
	Total	-	8/8	-	-	2/8	-
Isenman [43]	*L. longbeachae*	2/99	107/107	10/10	-	44/107	-
	*L. pneumophila*	-	19/19	-	-	12/19	-
	Total	2/99	126/126	10/10	-	56/126	-
Lindsay [49]	LpSG1	4/4	4/4	4/4	-	1/3	-
	LpSG12	0/1	1/1	1/1	-	0/1	-
	Total	4/5	5/5	5/5	-	1/4	-
Murdoch [52]	LpSG1	-	Serum: 0/2 Urine: 1/2	-	1/2	2/2	-
	LpSG3	-	Serum: 0/1 Urine: 0/1	-	0/1	0/1	-
	LpSG4	-	Serum: 1/2 Urine: 2/2	-	0/1	0/1	-
	LpSG4/5	-	Serum: 1/1 Urine: 1/1	-	1/1	1/1	-
	LpSG5	-	Serum: 1/1 Urine: 0/1	-	1/1	0/1	-
	LpSG6	-	Serum: 0/1 Urine: 0/1	-	0/1	0/1	-
	LpSG7	-	Serum: 1/1 Urine: 1/1	-	1/1	1/1	-
	LpSG10/12	-	Serum: 0/1 Urine: 1/1	-	0/1	0/1	-
	LpSG12/13	-	Serum: 0/1 Urine: 0/1	-	0/1	0/1	-
	LpSG13	-	Serum: 0/1 Urine: 1/1	-	1/1	1/1	-
	*L. micdadei **	-	Serum: 4/10 Urine: 5/10	-	4/6	0/6	-
	*L. longbeachae **	-	Serum: 1/3 Urine: 0/3	-	2/2	0/2	-
	*L. jordanis ***	-	Serum: 3/3 Urine: 0/3	-	0/3	0/3	-
	*L. bozemanae ***	-	Serum: 1/1 Urine: 0/1	-	0/1	0/1	-
	*L. gormanii*	-	Serum: 0/1 Urine: 1/1	-	-	-	-
		-	Serum: 13/30 Urine: 13/30	-	11/23	5/23	-
Pasculle (admission) [53]	LpSG1	-	-	-	5/8	8/8	6/8
	LpSG4	-	-	-	2/2	2/2	2/2
	LpSG6	-	-	-	1/1	1/1	1/1
	*L. micdadei*	-	-	-	3/3	3/3	3/3
	Total	-	-	-	11/14	14/14	12/14
Pasculle (follow-up) [53]	LpSG1	-	-	-	8/8	8/8	8/8
	LpSG4	-	-	-	2/2	2/2	2/2
	LpSG6	-	-	-	1/1	1/1	1/1
	*L. micdadei*	-	-	-	3/3	3/3	3/3
	Total	-	-	-	14/14	14/14	14/14
Pouderoux (initial infection) [54]	LpSG1	9/10	-	-	-	9/10	-
	LpSG3	-	-	-	-	1/1	-
	LpSG8	-	-	-	-	1/1	-
	Total	9/10	-	-	-	11/12	-
Pouderoux (recurrent infection) [54]	LpSG1	-	7/10	-	-	8/10	-
	LpSG3	-	1/1	-	-	1/1	-
	LpSG2-6-12	-	1/1	-	-	1/1	-
	Total	-	9/12	-	-	10/12	-
Waller [21]	*L. pneumophila*	8/16	4/5	Acute: 10/16 Convalescent: 2/11	-	-	-
	*L. longbeachae*	-	3/6	Acute: 19/28 Convalescent: 9/30	-	-	-
	Total	8/16	7/11	Acute: 29/44 Convalescent: 11/41	-	-	-
Total	-	23/130	181/222	55/100	36/51	113/213	26/28
Frequency of positivity (%)	-	17.69	81.53	55.00	70.59	53.05	92.86

* These species were observed in a coinfection. ** These species were observed in a separate coinfection. ^a^ All cases outlined in this table are confirmed cases. Numerators are the number of positive tests, while denominators are the number of total tests applied to that species or serogroup. Pasculle et al. included data from initial diagnoses and repeat diagnostics conducted over 9 days of patient hospitalization [53]. Results from both datasets are listed separately. Slowly resolving Legionnaires’ disease cases from Pouderoux et al. are included in the initial infections, while reinfections/recurrent infections are outlined in recurrent infections [54]. Abbreviations: SG, serogroup; UAT, urinary antigen test; PCR, polymerase chain reaction; DFA, direct fluorescent antibody.

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
