# Peer review of "The Adequacy of Current Legionnaires’ Disease Diagnostic Practices in Capturing the Epidemiology of Clinically Relevant Legionella: A Scoping Review"

_pathogens, 2024, doi:10.3390/pathogens13100857_

Round 1

Reviewer 1 Report

Comments and Suggestions for Authors

It is increasingly being recognised that clinical cases of pneumonia that might be LD but are not SG1 may go undetected  because of limitations of the assays used.  This is being brought to light in countries where qPCR is being used instead of or to augment UAT testing. However, this is often highlighted by individual studies. What this paper does is bring together the evidence in a rigorous and well summarised review. The screening process has been thoroughly described and the results are well presented.

I have no specific suggestions for changes to the manuscript. Just a very small number of typographical changes.    

Comments on the Quality of English Language

Well presented and easy to read. Just a very small number of typographical errors to pick up at editorial

Author Response

Comments and Suggestions for Authors

  1. It is increasingly being recognised that clinical cases of pneumonia that might be LD but are not SG1 may go undetected  because of limitations of the assays used.  This is being brought to light in countries where qPCR is being used instead of or to augment UAT testing. However, this is often highlighted by individual studies. What this paper does is bring together the evidence in a rigorous and well summarised review. The screening process has been thoroughly described and the results are well presented. I have no specific suggestions for changes to the manuscript. Just a very small number of typographical changes.

Answer: Thank you for the kind comments.

Comments on the Quality of English Language

  1. Well presented and easy to read. Just a very small number of typographical errors to pick up at editorial

Answer: Thank you. We reviewed the entire paper to confirm there are no typos.

Reviewer 2 Report

Comments and Suggestions for Authors

The reviewers present a scoping review in which they thoroughly evaluate the literature on the use of various diagnostic modalities for Legionnaires' disease (LD) and the epidemiology of different Legionella spp. This comprehensive review effectively summarizes the diagnostic methods and their efficacy in diagnosing LD, while also identifying potential areas for future research. Please consider the following suggestions to improve the presentation:

-          In the Study Selection section, “comparative quantitation using multiple detection methods” is listed as an inclusion criterion. Please consider explaining this further. Furthermore, in Table 6, the seven studies are reported as providing a breakdown by-technique. If the other studies did not include this breakdown, please clarify how they met the inclusion criterion.

-          The inclusion criterion of “at least 5 cases of LD in the patient group” may have been necessary for practical reasons, but it could potentially limit the comprehensiveness of the review and introduce bias in study selection. Please consider noting this as a limitation.

-          The flowchart indicates that a significant number of studies (n=12) were excluded due to “Article not available”. Could you please elaborate on this? Were these studies published only as abstracts, or what steps were taken before determining that the articles were unavailable?

-          The flowchart shows nine studies were included in the technical analyses; while lines #221-223 state that seven studies qualified. Please consider reviewing and updating the numbers to ensure consistency, or providing an explanation for the discrepancy.

-          In Table 6, the reviewers pool the results from different studies. While these results are compelling, given that the review is a scoping review rather than a systematic one, it does not include risk of bias or quality assessments. This may make evaluating the pooled results challenging. Please consider either removing the pooled results or providing justification and explanation for their inclusion in the Methods section.

-          The statements in lines #270-271: “We found that NAT-based techniques outperform non-NAT-based ones based on their higher frequency of positivity (Table 6).” and lines #307-308: “Our search found that PCR and DNA probes were the most reliable, regardless of Legionella subgroup (Table 6).” suggest conclusions that may not be fully supported by the data presented, as no formal comparison of NAT-based versus non-NAT-based techniques was conducted. Please consider revising the language to better reflect the scope of the analysis. 

Author Response

Comments and Suggestions for Authors

  1. The reviewers present a scoping review in which they thoroughly evaluate the literature on the use of various diagnostic modalities for Legionnaires' disease (LD) and the epidemiology of different Legionella spp.This comprehensive review effectively summarizes the diagnostic methods and their efficacy in diagnosing LD, while also identifying potential areas for future research. Please consider the following suggestions to improve the presentation:

In the Study Selection section, “comparative quantitation using multiple detection methods” is listed as an inclusion criterion. Please consider explaining this further. Furthermore, in Table 6, the seven studies are reported as providing a breakdown by-technique. If the other studies did not include this breakdown, please clarify how they met the inclusion criterion.

Answer: Thanks for your suggestion. The reason for including “comparative quantitation using multiple detection methods” as inclusion criteria is because Legionella diagnosis is often made using a single test. By requiring multiple detection methods, the diagnosis is more reliable and it helps ensure that the included studies were capable of detecting multiple species of Legionella. Additionally, we use multiple detection methods so we have information required to do comparative analyses between techniques.

As for the second comment, thanks for your suggestion. Our study includes several analyses. The analysis in Table 6 required that articles provide a by-technique breakdown of their findings. However, this breakdown is not necessary for other analyses we performed, as they were concerned with the diagnostic result regardless of technique. More details were added into lines 165-168 for clarity.

  1. The inclusion criterion of “at least 5 cases of LD in the patient group” may have been necessary for practical reasons, but it could potentially limit the comprehensiveness of the review and introduce bias in study selection. Please consider noting this as a limitation.

Answer: The minimum number of casesin a patient group was primarily to weed out case reports with unique presentations that are not representative of the typical course of Legionella infection and/or articles where patients/samples were evaluated for Legionella but 0 cases were found. For our investigations, including these articles with only a few cases of a unique presentation would also bias our by-species analyses with studies that don’t represent the typical frequency of Legionella in pneumonia cases. In total, 30 studies were not included due to insufficient presence of Legionella; 15 were pneumonia studies where 0 cases of Legionella spp were detected. 13 studies were also excluded for other reasons, most frequently a lack of serogroup/species analysis or a lack of NAT-based diagnosis methods. The remaining 2 studies consisted of only 4 cases of Legionella in total. We have added an explanation behind this in the discussion (lines 345-361)

  1. The flowchart indicates that a significant number of studies (n=12) were excluded due to “Article not available”. Could you please elaborate on this? Were these studies published only as abstracts, or what steps were taken before determining that the articles were unavailable?

Answer: The unavailable articles were published either before the 2000s or in the early 2000s. While there are abstracts available online, as far as we can tell the full articles were never uploaded to the internet, leaving us unable to find copies to review them.

  1. The flowchart shows nine studies were included in the technical analyses; while lines #221-223 state that seven studies qualified. Please consider reviewing and updating the numbers to ensure consistency or providing an explanation for the discrepancy.

Answer: Thanks for this comment. The correct number was seven for the technical analysis depicted in Table 6, that was an error in the flowchart that we have fixed. However, the Legionella cases in the cumulative pneumonia analysis depicted in Table 4 had nine qualifying studies and was not included in the flowchart at all. This has been addressed, and the flowchart now includes the correct number of qualifying studies for both the analyses depicted in Tables 4 and 6.

  1. In Table 6, the reviewers pool the results from different studies. While these results are compelling, given that the review is a scoping review rather than a systematic one, it does not include risk of bias or quality assessments. This may make evaluating the pooled results challenging. Please consider either removing the pooled results or providing justification and explanation for their inclusion in the Methods section.

Answer: Thank you. Table 6 currently depicts a qualitative summary of the findings from the results. You are correct regarding the challenge of generating pooled results, which would require a quantitative measure and confidence intervals obtained by meta-regression analysis, which are not reported in Table 6. To avoid any confusion, we added the word qualitative summary to the title of the table.

  1. The statements in lines #270-271: “We found that NAT-based techniques outperform non-NAT-based ones based on their higher frequency of positivity (Table 6).” and lines #307-308: “Our search found that PCR and DNA probes were the most reliable, regardless of Legionella subgroup (Table 6).” suggest conclusions that may not be fully supported by the data presented, as no formal comparison of NAT-based versus non-NAT-based techniques was conducted. Please consider revising the language to better reflect the scope of the analysis. 

Answer: Thank you for bringing this up. To improve the clarity of this we have stated the actual finding “Based on our dataset, we found that NAT-based techniques had a higher frequency of Legionella positivity compared to non-NAT-based ones.”, and The information included in our dataset suggests that PCR and DNA probes had a higher frequency of positivity, regardless of Legionella subgroup (Table 6).”

Thanks again for your suggestions and comments.